# The Risk of Sarcoidosis Misdiagnosis and the Harmful Effect of Corticosteroids When the Disease Picture Is Incomplete

**DOI:** 10.3390/biomedicines11010175

**Published:** 2023-01-10

**Authors:** Raluca Ioana Arcana, Radu Crișan-Dabija, Andrei Tudor Cernomaz, Ioana Buculei, Alexandru Burlacu, Mihai Lucian Zabară, Antigona Carmen Trofor

**Affiliations:** 1Doctoral School, Faculty of Medicine, University of Medicine and Pharmacy “Grigore T. Popa”, 700115 Iasi, Romania; 2Clinical Hospital of Pulmonary Diseases, 700115 Iasi, Romania; 3Faculty of Medicine, University of Medicine and Pharmacy “Grigore T. Popa”, 700115 Iasi, Romania; 4Institute of Cardiovascular Diseases “Prof. Dr. George I.M. Georgescu”, 700503 Iasi, Romania

**Keywords:** sarcoidosis, misdiagnosis, corticosteroids, immunomodulatory agents

## Abstract

Sarcoidosis is a complex granulomatous disease of unknown etiology. Due to the heterogeneity of the disease, the diagnosis remains challenging in many cases, often at the physician’s discretion, requiring a thorough and complex investigation. Many other granulomatous diseases have the potential to mimic sarcoidosis, whether infectious, occupational, or autoimmune diseases and starting an unnecessary corticosteroid treatment can worsen the patient’s prognosis, leading to side effects that can be harder to treat than the actual disease.

## 1. Introduction

Sarcoidosis is a complex multisystemic granulomatous disease, with unknown etiology and variable clinical manifestations, which frequently manifests with thoracic (bilateral hilar lymphadenopathies and pulmonary infiltrates), ocular, and cutaneous involvement. The diagnosis of sarcoidosis requires the identification of noncaseated epithelioid granulomas in one or more organs along with the exclusion of other pathologies that could cause granulomatous lesions. However, because of the similarities with other pathologies, an incomplete picture of the disease can lead to errors in the diagnosis, which is why in this paper we propose a detailed review of the most common pathologies and their role in a possible misdiagnosis regarding sarcoidosis [1,2]. 

## 2. Incidence

The onset age of sarcoidosis shows two peaks of incidence, with the first in the 3rd decade of life and another one in the 5th decade. In terms of geographic distribution, studies have shown higher incidences in the Scandinavian countries, with an incidence in Sweden of 11.5 per 100,000, followed closely by the United States at around 8–11 per 100,000 and Canada at 6.8 per 100,000. Very low incidence was reported in countries in East Asia, such as South Korea with 0.5–1.3 per 100,000. 

Racial variation is also encountered. For example, the non-Hispanic American black population tends to have a higher incidence; among them, black women have the highest rates.

Regarding the variations between the sexes, the results are controversial, since some studies reported a higher incidence in women and some studies found no differences at all [3,4]. 

## 3. Key Elements in Diagnosing Pulmonary Sarcoidosis

Often, in clinical practice, the heterogeneity of the disease can lead to an erroneous diagnosis of sarcoidosis, often with a significant impact on the patient’s life. In the following, we propose to highlight a series of investigations that can help establish a diagnosis as well as the diagnostic errors that may occur.

In regards to the symptoms, sarcoidosis can present itself in multiple forms and in various combinations, from asymptomatic or clinically insidious forms to acute forms with multiple organ involvement [5]. The most frequently affected organ is the lung, with a proportion of up to 90% [6], but the respiratory symptomatology remains in most cases poor, so regardless of the context, other signs and symptoms of the other organs must be looked for. In the table below (Table 1), we show some of the most common signs and symptoms that can occur in sarcoidosis:

Considering the extremely varied clinical appearance that can lead to confusion and misdiagnosis, the patient suspected of sarcoidosis requires a rigorous evaluation from an anamnestic, clinical, biological, radiological, and histological point of view, often requiring interdisciplinary collaboration between a pulmonologist, cardiologist, dermatologist, ophthalmologist, and neurologist. It is wrong to base the diagnosis on the mere association of some clinical symptoms. A thorough anamnesis is important, as it can identify exposures to respiratory toxins in the work environment (silica dust, beryllium) or exposures to various substances or antigens that could cause interstitial lung disease [5]. Highlighting a positive epidemiological context for tuberculosis in the family or in close contact, even in the past, may point the diagnosis toward tuberculosis.

### 3.1. Biomarkers in Sarcoidosis

Laboratory tests are an integral part of the diagnostic process of suspected sarcoidosis. Although there are no standardized or definitive biological markers, a series of tests can guide the diagnosis (Table 2).

ACE levels in patients suspected of sarcoidosis can also be influenced by genetic variations and ACE inhibitor therapy. One study by d’Alessandro et al., who analyzed serum samples from patients diagnosed with sarcoidosis and taking ACE inhibitor therapy, found that serum ACE levels were lower in these patients compared to those not taking ACEIs [16].

### 3.2. Imagistic Investigation in Pulmonary Sarcoidosis

Chest radiography has long been, and still is, a key investigation in the diagnosis of sarcoidosis. The Scadding classification made in 1961 divided sarcoidosis into five stages (Table 3) [5]. This classification is still used today, but the accuracy based on radiological presentation is only 50% [17]

On the other hand, the use of HRCT (high-resolution computer tomography) examination can detect abnormalities unidentifiable on a simple chest X-ray, including micronodules or pleural involvement, and can more accurately examine lung parenchyma, lung hilum, and mediastinal lymph node stations [17]. Considering the fact that chest HRCT examination is clearly superior to chest X-ray, it is useful to recommend and perform in the suite of investigations necessary to establish the diagnosis (Table 4).

One interesting imaging technique that can be useful in the diagnosis of sarcoidosis is 18 fluorodeoxyglucose (FDG) PET/CT. In thoracic sarcoidosis, mediastinal and hilar lymphadenopathy is found to be hypermetabolic on FDG PET/CT, as well as other parenchymal lesions such as nodules or masses. Although it is not recommended as a part of the initial workup, FDG PET/CT can provide essential information regarding the extent of the disease, cardiac and bone involvement, and the response to treatment, and can also guide a convenient biopsy site [22].

### 3.3. Fiberbronchoscopic Examination and Bronchoalveolar Lavage

Fiberbronchoscopic examination and bronchoalveolar lavage represent an integral part of the diagnosis of sarcoidosis and should be performed in every eligible patient who has no absolute contraindications.

Although in most cases the macroscopic endobronchial appearance is normal, the presence of granulations of the bronchial mucosa, secondary to granulomatous inflammation, erythema of the mucosa, and a cobblestone appearance, have been cited [7,23].

Bronchoalveolar lavage cytology is sometimes used as a biological marker for sarcoidosis, but its exact role is not yet clarified. Although lymphocytosis and normal neutrophil and eosinophil levels are frequent in the lavage of sarcoidosis patients, the diagnostic value of such findings is under debate; a similar situation is reported for CD4/D8 ratio [24]. Bronchoalveolar lavage cytology may reveal a moderate increase in the total number of cells, with lymphocytosis values between 30–50%. Although the evidence of increased neutrophils is rare in sarcoidosis, in cases with advanced pulmonary fibrosis, neutrophilia in the lavage can be identified. The CD4/CD8 ratio is frequently increased, and the cutoff used is a ratio value above 3.5. However, these values are more frequently found in the active stage of the disease. Normal CD4/CD8 ratio values do not exclude sarcoidosis, as the disease may be inactive at the time of examination [1].

On the other hand, lymphocytosis is not specific to sarcoidosis, being found in many other diseases such as tuberculosis or pulmonary mycosis, especially in the context of a CD4/CD8 ratio below 3.5 [25].

The advantage of the fiberbronchoscopic examination is the performance of biopsies using the different techniques illustrated, alone or combined (Table 5). The diagnostic yield is higher the more biopsy techniques are used [25]:

### 3.4. Histopathological Findings

The typical histological pattern is epithelioid granuloma with a compact agglomeration of epithelioid cells and multinucleated histiocytes, with rare lymphocytes, non-necrotizing. In rare cases, granulomas with caseification can also be found. However, highlighting the granuloma without caseification does not establish the diagnosis of sarcoidosis, as it is necessary to integrate clinical–radiological and serological investigations to form a complete picture, in order to attain a diagnosis of increased accuracy [5,9,25].

Even if the disease in question is pulmonary sarcoidosis, radioclinical manifestations are not sufficient for an accurate diagnosis, and obtaining biopsies from mediastinal/hilar lymphadenopathies or from the lesions expressed in lung parenchyma should represent a priority for the clinician to avoid a false diagnosis of sarcoidosis [25]. If obtaining material for the histopathological examination cannot be achieved through fiber bronchoscopy, mediastinoscopy or video-assisted lung biopsy may be indicated procedures if the patient does not have major contraindications [26].

Whenever possible, other organs should be investigated where a less invasive biopsy sample could be obtained, such as a skin lesion or a conjunctival nodule (with the exception of erythema nodosum lesions) [26].

The only form of sarcoidosis considered specific enough to not require a biopsy is Lofgren’s syndrome with the association of bilateral hilar adenopathy, erythema nodosum, fever and joint involvement, lupus pernio and Heerfordt syndrome [9,26].

## 4. Possible Errors in the Diagnosis of Sarcoidosis

The certainty diagnosis of sarcoidosis remains, most of the time, a diagnosis of exclusion, never certain, and often at the discretion of the attending physician. The nonspecific symptomatology, the granulomatous-type inflammation, that is also found in other pathological conditions, the absence of specific and sensitive markers and the lack of a perfect investigation, often make the diagnosis difficult and prone to errors, both through overdiagnosis and also by classifying it as another disease [5]. The result of these errors is reflected in the patient’s experience through incorrect management and treatment, which can aggravate the suffering and worsen the course of the underlying disease. 

Although the differential diagnosis of pulmonary sarcoidosis is exhaustive and includes both rare and frequent diseases, in the following we will discuss the most common pathologies that can mimic pulmonary sarcoidosis: respiratory tuberculosis, malignancy, pneumoconiosis, sarcoid-like reactions induced by drugs (DISR), hypersensitivity pneumonitis vasculitis and connective tissue diseases, and common variable immunodeficiency syndrome (CVID).

### 4.1. Respiratory Tuberculosis

Infection with Mycobacterium tuberculosis is the most common pulmonary infection of the granulomatous type. Not only in countries with a high incidence, but pulmonary tuberculosis should also represent the main differential diagnosis with sarcoidosis, as its wrong diagnosis can have nefarious consequences for patients and their peers.

From a clinical point of view, there is an undeniable similarity between sarcoidosis and pulmonary tuberculosis, which can make them, in the first phase, impossible to differentiate from one another. The insidious onset and nonspecific symptoms such as cough, fever, night sweats, weight loss, and fatigue are present in a large proportion of patients with both pathologies [4,26,27].

Although the radiological picture in tuberculosis is quite suggestive, with nodules, infiltrates, and a tendency to cavitation predominantly in the apical regions of the lung, there are situations in which the radiological presentation can be prone to confusion, especially in an immunosuppressed patient of whose radiological aspects tend to be nonspecific. On the other hand, the disseminated form of tuberculosis (miliary tuberculosis) can lead to a misdiagnosis of sarcoidosis, especially in the conditions of repeatedly negative bacteriological examination [4]. The presence of hilar and mediastinal lymphadenopathy, a pulmonary nodule or infiltrate with possibly pleural involvement, the presence of erythema nodosum, and ocular involvement can represent both sarcoidosis and primary tuberculosis [4].

Typical for tuberculosis is the histological evidence of granuloma with caseification necrosis. On the other hand, cases of tuberculosis with the presence of granuloma without caseification are cited in the literature and can occur in up to 20% of cases [26].

The clinical presentation of tuberculosis can be varied and atypical, as suggested by a case study by Chokoeva et al. that raised issues regarding the differential diagnosis in the case of a 35-year-old patient with skin lesions (erythematous plaques with asymmetric distribution on the face), bilateral cervical adenopathy, bilateral hilar adenopathy, and micronodules with bilateral distribution predominantly in the middle-lower lung areas, in which the histopathological exam highlighted sarcoid-like granulomas, but with a positive interferon-gamma release assays (IGRAs) test. A Ziehl–Neelsen staining was performed and found positive, which concluded the diagnosis as tuberculosis [28]. 

Extrapulmonary involvement should not exclude the diagnosis of tuberculosis, even in the context of the evidence of granulomas without caseification necrosis. Usama et al. reported a case concerning a 37-year-old patient with a history of chronic alcohol consumption, hospitalized for ascites of unknown etiology, in which the histopathological examination of a liver biopsy revealed granulomas without caseification necrosis, establishing the diagnosis of sarcoidosis. The evolution of the patient was unfavorable, which is why the investigations were continued and the CT examination revealed micronodular bilateral lung lesions, and the Ziehl–Neelsen staining from the sputum sample was found to be positive [29].

What the aforementioned suggests is the fact that the histopathological evidence may sometimes not be obvious for tuberculosis, but a complete set of investigations, including specific staining for the samples obtained (Ziehl–Neelsen, auramine–rhodamine), should be included in the management of patients in which the differential diagnosis between tuberculosis and sarcoidosis is not clear.

### 4.2. Malignancy and Sarcoidosis

Malignant conditions remain a diagnosis that can induce diagnostic errors with sarcoidosis, whether it is malignant solid tumors or lymphomas. Hodgkin’s and non-Hodgkin’s lymphomas can most frequently be confused with sarcoidosis, causing granulomatous reactions [4]. Hilar lymphadenopathy is encountered both in lymphoma and sarcoidosis, although unilateral or asymmetrical lymph node enlargement should favor the diagnosis of lymphoma [19]. The presence of constitutional symptoms and extrathoracic involvement such as skin, liver, and spleen involvement can be very similar to sarcoidosis. Moreover, ACE was found to be elevated in some patients with Hodgkin’s lymphoma, which tends to complicate the diagnosis even further [30].

Malignancy and sarcoidosis can have many overlapping features, making an accurate diagnosis difficult to establish. However, the association between sarcoidosis and cancer should be taken into account, as illustrated by Serra et al., who reported the case of a 65-year-old woman with a wrist mass, whose histopathological exam revealed epithelioid noncaseating granulomas, establishing the diagnosis of soft tissue sarcoidosis. The patient underwent further examination, which revealed intrathoracic lymphadenopathies and interstitial lung lesions, lesions that were biopsied, and confirmed lung sarcoidosis. Blood tests showed monoclonal IgG kappa gammapathy, and a bone marrow biopsy was performed, showing hypercellularity with 60% plasma cells and plasmocyte infiltration, establishing the diagnosis of multiple myeloma and sarcoidosis. In addition, the authors found another 33 case reports of both sarcoidosis and multiple myeloma [31].

Sarcoid-like reactions (SLRs) can be associated with various malignancies. The problem that such reactions pose is that the PET-CT examination cannot differentiate between sarcoid-like reactions and malignancy, because positive results are found in both situations. Unfortunately, no specific markers and no specific imagistic pattern were found for either of the two entities to be able to distinguish between them. Most of the time, invasive diagnostic methods are required, when obtaining histological material through fiberbronchoscopic methods is not possible [26,32].

### 4.3. Pneumoconiosis

Pneumoconiosis represents an occupational disease that occurs as a result of long and repeated exposure to various particles that induce a granulomatous pulmonary reaction.

Silicosis occurs as a result of prolonged exposure and repeated inhalation of silica dust, which causes an immune reaction in the lung parenchyma. The symptomatology is nonspecific and the radiological image can highlight nodular changes with perilymphatic distribution, areas of fibrosis, lymphadenopathies with or without areas of calcification, eggshell-like calcifications, and changes that may have a predilection for the upper lung regions. All these aspects can also be found in sarcoidosis, which makes the patient’s history a key element in identifying a professional exposure to silica dust [19].

Another type of pneumoconiosis that can lead to diagnostic confusion with sarcoidosis is berylliosis, an occupational disease that occurs secondary to prolonged exposure to beryllium and that can be difficult to differentiate radiologically, clinically, and histologically from sarcoidosis. There are many industries in which exposure to beryllium can occur, among the most common being the automobile industry, the biomedical industry, and the telecommunications industry [4].

Berylliosis can be misdiagnosed as sarcoidosis in many cases, either because of an incomplete history or simply because the patient or the clinician is unaware of the chronic exposure to beryllium. The patient’s professional history takes a primary role, and in situations where the clinician is not sure of the exposure, a thorough investigation can be carried out in order to obtain correct exposure information [4].

A study by Fireman et al. evaluated 47 patients reviewing their professional history, who were initially diagnosed with sarcoidosis. 14 of them had exposure to beryllium, and 3 patients out of the 14, in addition to the exposure history, also had a positive BeLTT test, being incorrectly diagnosed with sarcoidosis [33].

A case report that emphasized the importance of professional history is the one reported by Cheva et al., which presented the case of a 71-year-old patient with mediastinal adenopathy and bilateral pleural effusion, in which the histopathological examination of the biopsy sample obtained from the paratracheal adenopathy revealed the presence of epithelioid granulomas without caseification necrosis, with histiocytes and Langhans multinucleated giant cells, suggesting an inflammatory condition. During a detailed anamnesis, exposure to beryllium was identified, as he was a worker in the fertilizer industry and the established diagnosis was that of berylliosis [34].

### 4.4. Sarcoid-like Reactions Induced by Drugs (DISR)

Drug-induced sarcoid-like reactions—DISRs—refer to the development of a systemic granulomatous reaction, in direct relation to the administration of certain drugs. From a clinical, paraclinical, radiological, and histological point of view, it is similar to sarcoidosis, being almost impossible to differentiate in clinical practice. Bilateral hilar adenopathy, the presence of skin lesions, uveitis, and laboratory tests such as hypercalcemia and increased ACE in the serum represent elements that can be found in the picture of both conditions [4,35]. From a histopathological point of view, epithelioid granuloma without caseification necrosis surrounded by lymphocytes is found both in DISR and sarcoidosis [35].

The most frequently involved drugs are represented by immune checkpoint inhibitors, TNF-alpha antagonists, interferon, and antiretroviral therapy. In these situations, the patient’s history is very important, since these reactions can appear from 4 to 24 months after the start of the treatment, so each patient must be asked about the associated comorbidities and the treatment they are following. Stopping the treatment leads, in most cases, to the resolution of the syndrome. If DISR does not remit after stopping the offending drug, it can be treated similarly to sarcoidosis [4].

### 4.5. Hypersensitivity Pneumonitis

Hypersensitivity pneumonitis represents a group of pathologies that appear as a result of the inhalation of small organic particles and that induces a type 3 allergic reaction, with the formation of antigen–antibody complexes, and a type 4 reaction with a granulomatous reaction. The etiology of hypersensitivity pneumonitis is varied because multiple types of particles can determine the development of hypersensitivity pneumonitis: Farmer’s lung disease—exposure to moldy hay or wheat; Detergent lung disease—exposure to enzymes from the detergent industry; and Bird Fancier’s Lung are only a few suggestive examples [4,23].

From an imaging point of view, hypersensitivity pneumonitis and sarcoidosis can be similar by the distribution of nodular lesions or fibrotic changes predominantly in the upper lung regions. The nodules in sarcoidosis tend to be distributed perilymphatically, while in HP, the centrilobular distribution and ground-glass density represent the predominant imagistic features. The lack of lymphadenopathies should point the diagnosis toward hypersensitivity pneumonitis [19]. The histopathological examination showing granuloma without caseification necrosis can be misleading and can lead to a misdiagnosis of sarcoidosis. However, the granulomatous lesion tends to be small, poorly represented, and poorly demarcated, being more frequently associated with multinucleated giant cells [19].

Bronchoalveolar lavage can reveal lymphocytosis in both pathologies, but a study by Raghu et al. highlighted a higher lymphocytosis among HP patients. The CD4/CD8 ratio can represent a useful marker in the differential diagnosis, although sometimes the lavage results are unremarkable in both diseases [4,36].

The patient’s history of exposure to organic particles, and the evidence of specific serum immunoglobulins, in the context of a suggestive radioclinical picture, advocate for the diagnosis of hypersensitivity pneumonitis [36]. However, diagnosis can present difficulties when the source of exposure cannot be identified, or when the source of exposure is unknown to the patient.

### 4.6. Vasculitis and Connective Tissue Diseases

Autoimmune diseases should always be taken into consideration in the differential diagnosis of sarcoidosis. Patients who present with systemic, nonspecific symptoms and usually display a CT pattern of interstitial lung disease should be screened for systemic sclerosis, dermatomyositis, antisynthetase syndrome, and rheumatoid arthritis. The study of specific antibodies should not be skipped as it can add valuable information [37].

In regard to connective tissue diseases, the patient’s clinic may present elements such as nonspecific symptomatology (fever, fatigue, weight loss, night sweats) and extrapulmonary involvement, including skin and eye lesions. The presence of granulomatous lesions in the histological examination can complicate the diagnosis among these patients [38].

Vasculitis belongs to a group of autoimmune diseases characterized by the presence of granulomatous inflammation. They can resemble sarcoidosis from a clinical point of view, leading to diagnostic errors. When patients exhibit nonspecific symptoms, pulmonary involvement in association with renal dysfunction, vasculitis should be considered and the patient should undergo further investigations, as vasculitis can have a poor prognosis if untreated. From a histological point of view, there are differences that can orient the diagnosis toward one of these pathologies. The study of p-ANCA and c-ANCA antibodies represents an important step in the evaluation of the patient when there are diagnostic difficulties [4,37].

A case report by Thaniyavarn T. showcased the case of a 47-year-old man who was diagnosed with pulmonary sarcoidosis and was under treatment with infliximab. The initial diagnosis was established based on symptoms, with the negative serology for autoimmune diseases, suggestive lung imaging, and a Gallium scan showing uptake in both lungs, despite a biopsy and bronchoalveolar lavage not compatible with sarcoidosis. However, a decision was made to reevaluate the patient and repeat the serological tests, which showed CPK elevation and positive anti-PL-12, changing the diagnosis to antisynthetase syndrome [39].

### 4.7. Common Variable Immunodeficiency Syndrome (CVID)

Common variable immunodeficiency syndrome is a heterogeneous disease that represents the most frequent form of primary immunodeficiency. Its key elements are an increased susceptibility to recurrent infections, as well as the presence of hypogammaglobulinemia [40].

In some cases, CVID can be associated with granulomatous lymphocytic interstitial lung disease, with similar aspects to those encountered in sarcoidosis. Often, the symptoms are nonspecific, which can lead to errors in the diagnosis. However, the presence of splenomegaly and hypogammaglobulinemia, in addition to interstitial lung disease and a history of recurrent infections, should orientate the clinician toward common variable immunodeficiency syndrome, as misdiagnosis and treatment of CVID with corticosteroids can have a negative impact on these patients [32].

A study by Jamal et al. described the case of a 53-year-old patient admitted for supraclavicular adenopathy, weight loss, and asthenia. The laboratory tests showed isolated lymphopenia, slight hypercalcemia, normal serum protein electrophoresis, and elevated ACE and lysozyme. Imaging investigations revealed supraclavicular lymphadenopathy, mediastinal and hilar lymphadenopathy, pulmonary infiltrates, bibasal reticulation, ground-glass opacities, and bronchiectasis. BAL showed a CD4/CD8 ratio of 5.75 and the bronchial biopsies revealed noncaseating epithelioid and giant cellular inflammation. The diagnosis established was systemic sarcoidosis and the patient was placed under corticosteroids with an initial favorable clinical course. The treatment was stopped after a year. The serum protein electrophoresis was performed, and revealed persistent hypogammaglobulinemia with lymphocyte immunophenotyping, confirming the diagnosis of CVID [41].

## 5. Sarcoidosis Treatment

Not all patients with sarcoidosis require systemic treatment because sarcoidosis can evolve in two different ways: a time-limited one (in this case, two-thirds of the patients will experience a self-remitting disease) and a chronic one (in this case, 10–30% of patients require prolonged treatment). Treatment is usually reserved for life-threatening organ involvement (central nervous system sarcoidosis, portal hypertension, cardiac sarcoidosis, pulmonary hypertension, or advanced pulmonary fibrosis, etc.) or functional threat (laryngeal involvement and/or posterior uveitis, severe or defacing skin disease, etc.) [9].

Sarcoidosis treatment is not an easy task, this disease evolution can vary very much from asymptomatic forms with only radiographic abnormalities found on routine check-ups to more fulminant evolution that can result in pulmonary disability and death. Therefore, some patients are overtreated and others are undertreated. Before the decision to apply treatment, it is necessary to establish the risk versus the benefit of the treatment, so physicians must consider the minimization of the risk of disability and the risk of death due to the loss of lung function or quality of life (QoL) impairment and the risk of the occurrence of adverse effects of glucocorticoids, such as comorbidities or QoL loss [42]. The decision to treat pulmonary sarcoidosis can be made using some key patients assessments: risk of significant morbidity or mortality (danger), symptoms/ quality of life/ functional impairment, the likelihood of active granulomatous inflammation, and pulmonary physiology reduction. The decision to withhold treatment is imposed by the high likelihood of remission and risk of treatment toxicity.

Based on ERS clinical practice guidelines published in 2021, treatment with glucocorticosteroids should be implemented for patients that did not receive any treatment and have major involvement from pulmonary sarcoidosis in order to improve and/or preserve FVC and QoL [42]. The recommendation for pulmonary sarcoidosis is to use an initial dose of 20–40 mg/day prednisone, but the preferred dose among experts in this field is 20 mg, which should be reduced gradually over a period of 1 to 6 months to the lowest effective maintenance dose (≤10 mg/day) [43]. The usage of glucocorticoid-sparing agents, in this case, did not show promising results, but using high-dose inhaled glucocorticoids after treatment with oral GC for 3 months seems to decrease the risk of relapse and lower the need for long-duration treatment with CG. Studies have shown that methotrexate reduces the need of prednisone in patients suffering from acute sarcoidosis, and this diminishes weight gain [44]. 

Chronic pulmonary sarcoidosis is defined as an active disease for at least two years. Because of the known side effects of GC, the treatment of chronic pulmonary sarcoidosis should include second or third-line medications. The decision to use these lines of therapies should not be based on the duration of GC therapy, but rather on the significant risk of glucocorticoid toxicity or a clinical failure of glucocorticoid therapy. Second-line agents for pulmonary sarcoidosis include methotrexate (MTX), azathioprine, leflunomide, and mycophenolate. Third-line agents for pulmonary sarcoidosis include infliximab and adalimumab [43].

Key consensus points of the Delphi consensus recommendations for a treatment algorithm in pulmonary sarcoidosis stated that glucocorticoids should be used as initial therapy for most patients and that immunomodulators, usual methotrexate, should be considered in severe or extrapulmonary diseases that need a long period of treatment, or that they should be considered in patients with a high risk of steroid toxicity as a steroid-sparing intervention. Regarding biological therapies, the consensus recommendations state that this type of therapy can be added if nonbiologic therapies are insufficiently effective or are not tolerated with initial biologic therapy, usually with a tumor necrosis factor-α inhibitor, typically infliximab [45].

Treatment of extrapulmonary sarcoidosis should rely on three goals: improvement of organ dysfunction risks, mortality reduction, and quality of life improvement. The number of studies conducted in this field is limited. Before applying various treatments, some important points need to be considered: treatment should target the granulomatous process (anti-inflammatory drugs), and organ-directed treatments and supportive treatments should also be used. The treatment of organ dysfunctions can improve the outcome of sarcoidosis and include nonpharmacological therapies (pacemakers, CSF diversion to manage hydrocephalus, and transplantation for the heart, kidney, and liver) or pharmacological therapies (hormonal substitution, antiepileptic drugs or psychiatric medication, etc.) [46].

Experts in this field seem to agree that the best management of this type of patient should involve disease monitoring, and active treatment should depend on the tolerability of symptoms and risk of serious organ dysfunction. When prescribing the treatment, special attention should be given to patient-specific factors: In patients that are overweight and/or suffer from diabetes, GC should be avoided. Chronic kidney disease is a contraindication for treatment with methotrexate. Thiopurine S-methyltransferase (TPMT) deficiency is a contraindication for the use of azathioprine [47].

## 6. Harmful Effects of Corticosteroids

The intensity of the inflammation and the affected organ are two factors involved in the type of clinical manifestations of sarcoidosis. Because there are no curative treatments, only suppressive medications can be used, and this type of treatment can cause side effects. Guidelines regarding the treatment of sarcoidosis still recommend the use of systemic corticosteroids as first-line therapy, with a slow reduction in dosage to long-term low-dose administration. This type of dosing is used because of the toxicity of long-term use of systemic steroids and the belief that by using small doses, the impact will be less serious, but data published in the literature seem to show that this type of strategy is not sufficient because side effects are still reported: a 30% increase in hypertension, 20–30% prevalence of fracture or osteoporosis, and an increase of up to 4 times in hyperglycemia and type 2 diabetes. 

In a retrospective, matched cohort study conducted in 2022, Einarsdottir et al. investigate all-cause and disease-specific mortality in a large group of Swedish oral glucocorticoids (GC) users. The Swedish Prescribed Drug Registry and the Swedish Cause-of-Death Registry were used to obtain information about dispensed prescriptions and the cause of death. The study included patients that received treatment with prednisolone ≥5 mg/day (or equivalent dose of other GC) for ≥21 days between 2007–2014. Patients receiving treatment with prednisolone had significantly higher all-cause mortality compared to controls. The highest adjusted hazard ratio was observed in high-dose users for deaths from sepsis–6.71 (5.12 to 8.81) and pulmonary embolism—7.83 (5.71 to 10.74) [48].

Drent et al. conducted a study with the aim of assessing the prevalence of self-reported gastrointestinal side effects of these drugs. A total of 70% of the participants received one or more drugs, and the most prevalent and important side effect was weight gain caused by increased appetite among prednisone users [49]. In 2021, Vivienne Kahlmann et al. published the results of a study that investigated patient-reported side effects of prednisolone and methotrexate. A total of 67 patients were asked to complete a questionnaire about medication use and 58 were included in the present study (the 8 excluded patients never used prednisolone or methotrexate); 89% were using or used prednisolone and 70% methotrexate. In the case of patients using prednisone, 78% reported at least one side effect (48%—weight gain, 24%—psychological problems/behavior change, 20%—sleep problems/fatigue). In the group of patients that used methotrexate, the side effects reported were fewer: only 49% reported one or more side effects (31%—nausea or other gastrointestinal complaints, 10%—general malaise, 5%—headache, 5%—liver test abnormalities, and 5%—hair loss). The results of this study show that prednisone is linked to more side effects than methotrexate and that the side effects of methotrexate, if any, are less bothersome [50].

Data in the literature show that treatment with GCs is associated with a decrease in bone mineral density and an increased fracture risk, depending on the daily dose. The fracture risk occurs rapidly and decreases quickly to baseline after treatment discontinuance [51,52,53]. A case-control study was conducted by Oshagbemi et al. assessing the risk of major osteoporotic fractures in patients suffering from sarcoidosis receiving treatment with glucocorticoids. With the aim of deriving adjusted odds ratios (OR), conditional logistic regression models were used. Two groups were formed: the first group was formed of a total of 376,858 subjects with a major osteoporotic fracture, and the second group was formed of the same number of subjects without major osteoporotic fracture. The results showed that in the 124 patients with sarcoidosis, the risk for major osteoporotic fracture was associated with the current use of GC (OR 1.74; 95% CI 1.17–2.58) and that the risk reached baseline after treatment interruption. In this type of patients, the cumulative dose of prednisolone equivalents of 1.0–4.9 g and more than 10 g were associated with a higher risk of major osteoporotic fracture. Moreover, the results show that in both groups, current exposure to GC is associated with an increased risk of major osteoporotic fractures: the risk of major osteoporotic fracture in the second group was OR 1.36; 95% CI 1.32–1.40. Sarcoidosis per se is not associated with increased fracture risk [54]. The association between sarcoidosis and increased calcium metabolism is well known; there are concerns about vitamin D and calcium supplementation, as hypercalcemia and calciuria may have serious consequences. Still, there are data supporting the use of vitamin D and calcium supplements [55].

Cardiovascular risks of corticosteroids need to also be considered in the potential cardiovascular expression of sarcoidosis, and further research is needed to document the optimal dose and duration, or even to explore the benefits of corticosteroids administered via a safer inhalator route, when beneficial [56].

A cohort study that investigated the risk of developing Type 2 diabetes (T2D) in patients suffering from sarcoidosis and the implication of treatment with corticosteroids in increasing the risk was published by Entrop et al. The results show that the risk of T2D was higher for patients suffering from sarcoidosis and receiving corticosteroid treatment in the first 2 years after diagnosis (The HR for T2D was 1.4 (95% CI 1.2–1.8) associated with untreated sarcoidosis and 2.3 (95% CI 2.0–3.0) associated with corticosteroid-treated sarcoidosis) [57].

A group of international physician experts from eleven subspecialties (rheumatology, psychiatry, pediatric rheumatology, nephrology, dermatology, neurology, ophthalmology, infectious disease, and pulmonology) constructed an instrument that can measure glucocorticoid (GC) toxicity. The aim of the Glucocorticoid Toxicity Index (GTI) is to determine and measure the modifications that can occur in GC toxicity compared to points in time. Some of the domains investigated are body mass index, low-density lipoprotein concentrations, blood pressure, hemoglobin A1c, medication changes, occurrences of infections, assessments of muscle strength, skin toxicity, and the neuropsychiatric effects of glucocorticoids. GTI has been authorized by more than 45 studies, including 12 phase-3 clinical trials. At the present time, GTI can be used in clinical practice and clinical trials with the main aim of assessing the benefits of new medications that become available in treating this type of disease [58].

When a misdiagnosis occurs and treatment with corticosteroids starts, the side effects depend on the misdiagnosed disease. For example, the use of corticosteroids in treating pulmonary TB has been argued over time, and experts in the field have different opinions regarding the outcomes. Studies conducted in this field show that long-term use of corticosteroids can make the patient susceptible to mycobacterial infection and to the development of TB, as well as reactivate latent bacilli within macrophages [59,60]. The results of another study published in 2017 showed that mycobacterial survival is augmented by glucocorticoids through the inhibition of TBK1 Kinase and by downregulating genes promoting autophagy [61]. Considering that corticosteroids may reactivate latent TB infection and increase mycobacterial growth, as well as the lack of evidence for the impact that this type of treatment has on the mortality of this disease, the corticosteroid should be used only in special cases (extrapulmonary TB) and with caution [60]. 

Even though corticosteroids are used in many cancer treatment regimens, side effects appear and complicate their use. Neuropsychiatric toxicity is revealed by symptoms such as insomnia, cognitive impairment and mood symptoms, and severe mental disorders, including mania, psychosis, and severe depression [62]. Hyperglycemia is another adverse effect common in patients with hematologic malignancies treated with corticosteroids. Higher steroid doses and long-acting steroids cause a greater degree of hyperglycemia [63]. 

Alternatives to corticoids exist. Besides classic immunomodulators (such as methotrexate or mycophenolate), there are data suggesting a role for some of the available TNF inhibitors such as infliximab or adalimumab; furthermore, there is some hope regarding the use of anti-B agents such as rituximab. A more thorough understanding of inflammatory pathways is necessary in order to make use of contradictory reports regarding a plethora of other agents [64].

## 7. Conclusions

A complete picture of the disease is required to establish an accurate diagnosis. The clinician should always conduct a thorough investigation of the patient, checking all the points, from simple anamnesis to complex investigations, in order to avoid a misdiagnosis. As demonstrated above, quite a few diseases can be misinterpreted as sarcoidosis, which can lead to inappropriate treatment, poor prognosis, and low quality of life.

Despite a long history, sarcoidosis remains a challenge for everyday clinical practice. Initial diagnosis, decision to treat or to stop treating, choosing an immunomodulatory agent, or managing an unresponsive case is still difficult, despite existing management protocols. Corticosteroids are still considered first-line agents despite increasing data on their detrimental side effects. Patient education, comorbidity assessment, and side effect monitoring are paramount to avoid such unfavorable outcomes. 

## Figures and Tables

**Table 1 biomedicines-11-00175-t001:** Organ involvement and symptoms of sarcoidosis [6,7,8,9,10,11].

General Symptoms	Fever, Weight Loss, Fatigue, Concentration Disturbances, Night Sweats
Thoracic manifestations Over 90% of cases	Cough, chronic dyspnoea, chest pain, wheezing Bilateral hilar/mediastinal lymphadenopathy Pulmonary infiltrates
Ocular involvement 10–25% of cases	Anterior or posterior uveitis, conjunctivitis, sicca syndrome, scleritis, episcleritis
Skin lesions Up to 1/3 of the cases	Erythema nodosum, lupus pernio, granuloma development on scars
Cardiac involvement 1–23% of cases	Can be clinically occult or Chest pain, palpitations, dyspnoea, syncope Arrythmias—total heart block, tachyarrhythmias, cardiac failure
Musculoskeletal involvement 5–15% of cases for joint involvement	Sarcoid arthropathy, sarcoid myopathy, chronic sarcoid arthritis
Renal involvement Up to 20% of cases	Granulomatous interstitial nephritis, nephrocalcinosis, nephrolithiasis
Neurologic involvement 3–10% of cases	Involvement of the cranial nerves: II, VII, VIII Central nervous system or peripheral nervous system involvement Meningeal irritation Symmetrical mono/polyneuropathies, polyradiculopathy Affecting the pituitary gland—diabetes insipidus, amenorrhea, galactorrhea
Otolaryngological involvement 5–15% of cases	Sinonasal involvement—nasal obstruction, rhinitis, epistaxis, rhinorrhea, anosmia Laryngeal sarcoidosis—hoarseness, dysphagia, inspiratory dyspnoea
Löfgren Syndrome	Acute onset with fever, bilateral gilar lymphadenopathy, erythema nodosum, joint involvement
Heerfordt Syndrome	Fever, enlargement of the salivary/parotid glands, facial nerve paralysis, anterior uveitis

**Table 2 biomedicines-11-00175-t002:** Serum biomarkers in sarcoidosis [1,5,12,13,14,15].

Serum angiotensin convertase—ACE	Enzyme produced by epithelioid cells derived from activated macrophages from the site of the granuloma Increased values in the serum of patients with sarcoidosis, especially in active forms A normal value does not necessarily exclude sarcoidosis ACE can also be found increased in other granulomatous diseases, such as tuberculosis and pneumoconiosis Remains poorly specific and sensitive for the diagnosis of sarcoidosis
Serum soluble Interleukin 2 receptor—sIL-2R	Marker of Th1 cell activation in relation to the evolution of the granuloma Elevated levels of sIL-2R have been described in active sarcoidosis and multiple organ involvement There are a number of other diseases in which sIL-2R can be elevated, such as other granulomatous diseases, infections, lymphoproliferative diseases, and autoimmune disorders
Cytokines (IL-18, IL-12), neopterin, lysozyme, serum amyloid A	Often useful in patients for monitoring the evolution of the disease, less for diagnosis because they do not have a high enough accuracy Most of the time, these biomarkers are difficult and expensive to use in a clinical setting

**Table 3 biomedicines-11-00175-t003:** Radiological classification of pulmonary sarcoidosis [5].

Stage of Disease	Radiological Findings
Stage 0 disease	Normal Findings on Chest Radiography
Stage 1 disease	Bilateral hilar/mediastinal lymphadenopathy
Stage 2 disease	Bilateral hilar/mediastinal lymphadenopathy and pulmonary infiltrates
Stage 3 disease	Pulmonary infiltrates without hilar lymphadenopathy
Stage 4 disease	Pulmonary fibrosis

**Table 4 biomedicines-11-00175-t004:** HRCT aspects that can be encountered in sarcoidosis [18,19,20,21].

The Predominance of Lesions in the Middle and Upper Lung Regions	Suggestive of Sarcoidosis
Perilymphatic micronodules between 1–5 mm, most frequently located subpleural, along the interlobular septae or interlobar fissures, respectively, of the bronchovascular bundles	Suggestive of sarcoidosis
The galaxy sign (central dominant nodule surrounded by numerous satellite micronodules)	Lends itself to diagnostic confusion with malignant lung disease
Miliary pattern	Very rare cases
Symmetrical, bilateral adenopathies	Mediastinal and hilar lymph node involvement—frequently right lower paratracheal, right hilar, subcarinal, subaortic, interlobar stations Isolated mediastinal adenopathy—atypical
Fibrotic changes	Distortions of the normal lung parenchyma architecture, with reticular thickening, bronchiectasis and traction bronchiole ectasis, respectively, and loss of lung volume
Atypical patterns	Cavitary lesions, solid lesions surrounded by ground-glass opacities known as the halo sign or atoll sign—ground-glass opacity surrounded by a denser consolidation and pleural effusions

**Table 5 biomedicines-11-00175-t005:** Diagnostic biopsy yield in different fiberbronchoscopic techniques [25].

Method of Biopsy	Diagnostic Yield
Endobronchial biopsy (EBB)	Variable between 20–61% Higher accuracy when endobronchial abnormal appearances are present
Transbronchial lung biopsy (TBB)	Variable detection rate between 12–66% for Stage 1 disease
Conventional transbronchial needle aspiration (c-TBNA)	Variable depending on the lymph node sampling, between 6–90% diagnostic accuracy
Endobronchial ultrasound-guided transbronchial needle aspiration (EBUS—TBNA)	Variable between 80–94%
Transbronchial lung cryobiopsy (TBLC)	Variable between 74–98%, pooled estimate at 80%

## Data Availability

No new data were created or analyzed in this study. Data sharing is not applicable to this article.

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
