# Peer review of "The Risk of Sarcoidosis Misdiagnosis and the Harmful Effect of Corticosteroids When the Disease Picture Is Incomplete"

_biomedicines, 2023, doi:10.3390/biomedicines11010175_

Round 1

Reviewer 1 Report

In this narrative review, the authors describe sarcoidosis, focusing on organ involvement, biomarkers, differential diagnosis, and treatment.

Overall, I find the paper interesting. I have only a few comments.

Specific comments to address are:

Introduction

I miss a sentence at the end of the section, informing the reader about what to expect in the rest of the paper.

Line 29. Consider moving the incidence data to a separate section and adding the incidence frequency in the different counties.

Key elements…

Line 4, page 2. Table 1. Adding frequency to organ involvement would make the table more clinically relevant. Ex. Respiratory symptoms  90%, ocular involvement 20% …

Biomarkers in sarcoidosis

ACE as a biomarker: Not defined in line 7, page 3.

      ACE is also affected by the use of ACE inhibition (as in hypertension) and        genetic polymorphism.

IL2R is the second most used biomarker in sarcoidosis and should be mentioned more in the paper.

Line 11, page 3, HRCT not defined.

It is perhaps outside the scope of this article. Still, one could consider including a section on PETCT scanning, which is increasingly used in sarcoidosis, to determine organ involvement and disease activity.

Errors in the Diagnosis of sarcoidosis

From page 5 to page 10: Although interesting, this section is way too long, particularly TB and malignancy. 

Treatment

I think ERS latest guidelines should be mentioned as a reference. (Baughman et al. ERS clinical practice guidelines on treatment of sarcoidosis, European Respiratory Journal, 2021)

Page 11. Is Thalidomide recommended by ERS? Otherwise, consider removing it or at least reducing it.

Is it meant that the first row in the tables is in bold?

Typos:

Line 9, page 5: t

Line 33, page 6: tiwh

Line 41, page 12: -

Author Response

Dear Reviewer, thank you for your valuable comments.

We respond punctually below.

1. Introduction

I miss a sentence at the end of the section, informing the reader about what to expect in the rest of the paper.

A sentence was added with reference to what to expect in rest of the paper.

2. Line 29. Consider moving the incidence data to a separate section and adding the incidence frequency in the different counties.

Incidence was moved to another section and completed

3. Key elements…

Line 4, page 2. Table 1. Adding frequency to organ involvement would make the table more clinically relevant. Ex. Respiratory symptoms  90%, ocular involvement 20% …

Frequency per organ involvement was added

4. Biomarkers in sarcoidosis

ACE as a biomarker: Not defined in line 7, page 3.

ACE was defined

ACE is also affected by the use of ACE inhibition (as in hypertension) and genetic polymorphism.

A paragraph was added in regard to this idea.

IL2R is the second most used biomarker in sarcoidosis and should be mentioned more in the paper.

A paragraph was added in regards to this idea.

5. Line 11, page 3, HRCT not defined.

HRCT was defined

It is perhaps outside the scope of this article. Still, one could consider including a section on PETCT scanning, which is increasingly used in sarcoidosis, to determine organ involvement and disease activity.

A paragraph was added in regard to PETCT scanning.

6. Errors in the Diagnosis of sarcoidosis

From page 5 to page 10: Although interesting, this section is way too long, particularly TB and malignancy. 

The introduction was shortened, 2 paragraphs were removed from the malignancy section, one paragraph was removed from the TB section.

7. Treatment

I think ERS latest guidelines should be mentioned as a reference. (Baughman et al. ERS clinical practice guidelines on treatment of sarcoidosis, European Respiratory Journal, 2021)

ERS guidelines were mentioned as a reference

8. Page 11. Is Thalidomide recommended by ERS? Otherwise, consider removing it or at least reducing it.

The paragraph was removed because thalidomide is not recommended by ERS guideline

9. Is it meant that the first row in the tables is in bold?

The first row in the tables was corrected

10. Typos:

Line 9, page 5: t

Line 33, page 6: tiwh

Line 41, page 12: -

Typos were corrected

Reviewer 2 Report

This manuscript revises much of the current knowledge on sarcoidosis, focusing on differential diagnosis.

The manuscript is well written, could be a nice read, but there are some confusing or incorect parts that need to be addressed.

The title should be rephrased, as it is in itself a short story, but doesn't actually completely overlap with the content. I should suggest "The risk of sarcoidosis misdiagnosis"

Page 1 row 24: should refer to thoracic, not pulmonary sarcoidosis (as the lymph nodes are not lung)

If geographic differences are mentioned, also rasial variation should be, mentioning the high incidence in afro-americans (page 1, rows 30-33)

As the manuscript focuses on differential diagnosis, the part dedicated to positive diagnosis should be shortened, condensed, possibly by the use of more tables and less text.

Page 1 row 35: not clear what the term "triad of criteria" refers to, for diagnosis, and how does this make the diagnosis clear. It should be stated that the diagnosis is based on a concordant complex of clinical elements, investigations, hystology, long term observation and exclusion of other conditions.

Page 1 row 42: replace organ damage with organ involvement

Tables should be mentioned in text in brackets.

Table 1: Respiratory symptoms should be replaced with "thoracic manifestations"

In the entire document, according to context, "pathology" should be replaced with "condition", a better English term (this is an obvious mis-translation from Romanian).

Table 1: CNS and PNS should be explicited, at least at the bottom of table.

The section "Errors in the diagnosis of sarcoidosis" should be called Possible errors in diagnosis. It would be better to stress for each condition what are the points of confusion and what is the risk of an undue treatment in case of diagnostic error. The initial quite large introduction should be reduced, as it is redundant.

For tuberculosis, mind that IGRA tests are not useful for the diagnosis of active tuberculosis, and cannot be used to differentiate sarcoidosis from TB! (page 7, row 1).

Page 7 rows 7-8. The confounding element between sarcoidosis and lymphoma is not only the granulomatous reaction, but the presence of lymph node enlargement, increased ACE, lung and skin involvement. This section should be more developed.

Page 7 row 20:  "histological examination of granulomas cannot differentiate between malignancy and sarcoidosis" is a confusing statement. Does this refer to the sarcoid reaction that can accompany malignancy, or is it a true histologic differentiation between sarcoid and malignant lesions, that can be similar on imaging? This should be clarified. Sarcoid-like reactions are mentioned only further (row 35), apparently not connected to the statement on row 20.

Page 8 row 30: add "drug induced" to "sarcoid-like reaction"

Page 10, Sarcoidosis treatment. This section should start with the statement that not all sarcoidosis patients need treatment, as about two thirds of them will recover spontaneously.

Page 10, row 25. Here only the decision to treat pulmonary sarcoidosis is discussed. Actually, the risk of the disease refers also to extrathoracic involvement prone to severe evolution (heart, eye, nervous system, kidney), that are not mentioned here at all. Involvement of these delicate organs is an important part of the decision to treat sarcoidosis.

Page 10, row 31. Acute sarcoidosis typically refers to Loefgren syndrome, which does not require corticosteroid treatment. The term "acute" here is confounding, and does not match the typical terminology used in the literature to describe the evolution of the disease. The association between "acute sarcoidosis" and "preferred treatment are GC" is not correct. Actually, the entire paragraph should be rethinked and rephrased, as it is confusing.

Page 11, rows 10-30. The section about thalidomide should be shortened, as it is not so important in the general economy of the manuscript.

The Treatment section should mention the treatment for extrapulmonary sarcoidosis.

The section Harmful effects of corticosteroids is a general review (and extensive) of the corticosteroids side effects, but does not relate to the current title, which points to the harmful effect of treatment if the diagnosis is wrong. Like tuberculosis or cancer treated with corticosteroids. 

Author Response

Dear Reviewer, thank you for your valuable comments.

We respond punctually below.

This manuscript revises much of the current knowledge on sarcoidosis, focusing on differential diagnosis.

The manuscript is well written, could be a nice read, but there are some confusing or incorrect parts that need to be addressed.

The title should be rephrased, as it is in itself a short story, but doesn't actually completely overlap with the content. I should suggest "The risk of sarcoidosis misdiagnosis"

The title was modified

  1. Page 1 row 24: should refer to thoracic, not pulmonary sarcoidosis (as the lymph nodes are not lung)

Modified to “thoracic sarcoidosis” instead of “pulmonary sarcoidosis”.

  1. If geographic differences are mentioned, also rasial variation should be, mentioning the high incidence in afro-americans (page 1, rows 30-33)

The paragraph regarding incidence was completed.

  1. As the manuscript focuses on differential diagnosis, the part dedicated to positive diagnosis should be shortened, condensed, possibly by the use of more tables and less text.

The text was condensed, 2 more tables were added in the part dedicated to positive diagnosis.

  1. Page 1 row 35: not clear what the term "triad of criteria" refers to, for diagnosis, and how does this make the diagnosis clear. It should be stated that the diagnosis is based on a concordant complex of clinical elements, investigations, hystology, long term observation and exclusion of other conditions.

The sentence with triad of criteria. was removed due to the error.

  1. Page 1 row 42: replace organ damage with organ involvement

 Replaced organ damage with organ involvement.

  1. Tables should be mentioned in text in brackets.

Modified in text

  1. Table 1: Respiratory symptoms should be replaced with "thoracic manifestations"

Replaced respiratory symptoms with thoracic manifestations.

  1. In the entire document, according to context, "pathology" should be replaced with "condition", a better English term (this is an obvious mis-translation from Romanian).

Pathology was replaced with condition

  1. Table 1: CNS and PNS should be explicited, at least at the bottom of table.

CNS and PNS were defined in table 1

  1. The section "Errors in the diagnosis of sarcoidosis" should be called Possible errors in diagnosis. It would be better to stress for each condition what are the points of confusion and what is the risk of an undue treatment in case of diagnostic error. The initial quite large introduction should be reduced, as it is redundant.

The title was modified to possible errors in the diagnosis of sarcoidosis. The introduction was shortened.

  1. For tuberculosis, mind that IGRA tests are not useful for the diagnosis of active tuberculosis, and cannot be used to differentiate sarcoidosis from TB! (page 7, row 1).

The paragraph with IGRA tests was removed

  1. Page 7 rows 7-8. The confounding element between sarcoidosis and lymphoma is not only the granulomatous reaction, but the presence of lymph node enlargement, increased ACE, lung and skin involvement. This section should be more developed.

A paragraph regarding lymphoma and sarcoidosis was added.

  1. Page 7 row 20:  "histological examination of granulomas cannot differentiate between malignancy and sarcoidosis" is a confusing statement. Does this refer to the sarcoid reaction that can accompany malignancy, or is it a true histologic differentiation between sarcoid and malignant lesions, that can be similar on imaging? This should be clarified. Sarcoid-like reactions are mentioned only further (row 35), apparently not connected to the statement on row 20.

This paragraph was removed because of the confusion

  1. Page 8 row 30: add "drug induced" to "sarcoid-like reaction"

Drug induced was added to sarcoid like reactions

  1. Page 10, Sarcoidosis treatment. This section should start with the statement that not all sarcoidosis patients need treatment, as about two thirds of them will recover spontaneously.

The start of this section was modified as recommended

  1. Page 10, row 25. Here only the decision to treat pulmonary sarcoidosis is discussed. Actually, the risk of the disease refers also to extrathoracic involvement prone to severe evolution (heart, eye, nervous system, kidney), that are not mentioned here at all. Involvement of these delicate organs is an important part of the decision to treat sarcoidosis.

New information was added regarding this subject

  1. Page 10, row 31. Acute sarcoidosis typically refers to Loefgren syndrome, which does not require corticosteroid treatment. The term "acute" here is confounding, and does not match the typical terminology used in the literature to describe the evolution of the disease. The association between "acute sarcoidosis" and "preferred treatment are GC" is not correct. Actually, the entire paragraph should be rethinked and rephrased, as it is confusing.

Modification were made to the paragraph and the term `acute sarcoidosis` was removed

  1. Page 11, rows 10-30. The section about thalidomide should be shortened, as it is not so important in the general economy of the manuscript.

The section about thalidomide was removed

  1. The Treatment section should mention the treatment for extrapulmonary sarcoidosis.

Information about the treatment of extrapulmonary sarcoidosis was added

  1. The section Harmful effects of corticosteroids is a general review (and extensive) of the corticosteroids side effects, but does not relate to the current title, which points to the harmful effect of treatment if the diagnosis is wrong. Like tuberculosis or cancer treated with corticosteroids. 

Information about corticosteroids side effects in tuberculosis and cancer patients was added
